# SORTED EIGENVALUE COMPARISON $d_{\mathsf{Eig}}$: A SIMPLE ALTERNATIVE TO $d_{\mathsf{FID}}$

## ABSTRACT

For $i = 1, 2$, let $\boldsymbol{S}_i$ be the sample covariance of $\boldsymbol{Z}_i$ with $n_i$ $p$-dimensional vectors. First, we theoretically justify an improved Fréchet Inception Distance ($d_{\mathsf{FID}}$) algorithm that replaces np.trace(sqrtm($\boldsymbol{S}_1\boldsymbol{S}_2$)) with np.sqrt(eigvals($\boldsymbol{S}_1\boldsymbol{S}_2$)).sum(). With the appearance of unsorted eigenvalues in the improved $d_{\mathsf{FID}}$, we are then motivated to propose sorted eigenvalue comparison ($d_{\mathsf{Eig}}$) as a simple alternative: $d_{\mathsf{Eig}}(\boldsymbol{S}_1, \boldsymbol{S}_2)^2 = \sum_{j=1}^{p}(\sqrt{\lambda_j^1} - \sqrt{\lambda_j^2})^2$, and $\lambda_j^i$ is the $j$-th largest eigenvalue of $\boldsymbol{S}_i$. Second, we present two main takeaways for the improved $d_{\mathsf{FID}}$ and proposed $d_{\mathsf{Eig}}$. (i) $d_{\mathsf{FID}}$: The error bound for computing non-negative eigenvalues of diagonalizable $\boldsymbol{S}_1\boldsymbol{S}_2$ is reduced to $\mathcal{O}(\varepsilon)\|\boldsymbol{S}_1\|\|\boldsymbol{S}_1\boldsymbol{S}_2\|$, along with reducing the run time by $\sim 25\%$. (ii) $d_{\mathsf{Eig}}$: The error bound for computing non-negative eigenvalues of sample covariance $\boldsymbol{S}_i$ is further tightened to $\mathcal{O}(\varepsilon)\|\boldsymbol{S}_i\|$, with reducing $\sim 90\%$ run time. Taking a statistical viewpoint (random matrix theory) on $\boldsymbol{S}_i$, we illustrate the asymptotic stability of its largest eigenvalues, *i.e.*, rigidity estimates of $\mathcal{O}(n_i^{-\frac{1}{2}+\alpha})$. Last, we discuss limitations and future work for $d_{\mathsf{Eig}}$.

## 1 INTRODUCTION

```python
import numpy as np
from scipy.linalg import eigvals, eigvalsh

# The square of improved dFID
def dFID(mean1, cov1, mean2, cov2):
    eigval = eigvals(cov1 @ cov2)
    # Round computational errors (if exist)
    # that lead to negative eigenvalues close to 0
    eigval[eigval < 0] = 0
    dif = mean1 - mean2
    res = dif.dot(dif) + np.trace(cov1 + cov2)
    return res - 2 * np.sqrt(eigval).sum()

# The square of proposed dEig
def dEig(scm1, scm2):
    # Sorted eigenvalues
    eigval1 = eigvalsh(scm1)
    eigval1[eigval1 < 0] = 0
    eigval2 = eigvalsh(scm2)
    eigval2[eigval2 < 0] = 0
    dif = np.sqrt(eigval1) - np.sqrt(eigval2)
    return dif.dot(dif)
```

Figure 1: Python codes for the square of improved $d_{\mathsf{FID}}$ and proposed $d_{\mathsf{Eig}}$.

In the image domain, it is of great interest to analyze the distribution shift between two collections of data entries (Wiles et al., 2021; Borji, 2019). On one hand, this is driven by the increasing awareness about the violation of the assumption of 'identical distribution' between training and (real-world) test datasets (Wu et al., 2022b). As for instance illustrated in the leaderboard of WILDS (Koh et al., 2021; Sagawa et al., 2021), many algorithms suffer from performance degradation and fail to generalize to heterogeneous testing settings. On the other hand, the importance of assessing distribution shift has been recognized with the rise of generative adversarial nets (GAN) (Goodfellow et al., 2014; Heusel

et al., 2017). The rapid development of GAN variants (Karras et al., 2019; 2020b) urges reliable and accurate metric(s) to assess the discrepancy between generated and real images (Borji, 2019).

To objectively assess GAN models, researchers have proposed a plethora of evaluation scores including Inception Score (Salimans et al., 2016), Kernel Inception Distance ($d_{\mathsf{KID}}$) (Bińkowski et al., 2018), and Precision/Recall (Kynkäänniemi et al., 2019; Sajjadi et al., 2018) (please also see (Borji, 2019; 2022) for in-depth review). Among various scores, Fréchet Inception Distance ($d_{\mathsf{FID}}$) (Heusel et al., 2017) is arguably the most widely-used metric for benchmarking GAN performance (Parmar et al., 2022). This is mainly due to the favorable theoretical property of being a mathematical metric (Dowson & Landau, 1982) and practical property of being well-correlated with perceived image quality (Sajjadi et al., 2018). Meanwhile, Chong & Forsyth (2020) argued that $d_{\mathsf{FID}}$ is a biased estimator and Kynkäänniemi et al. (2022) observed its undesirable sensitivity towards fringe features or classes. Despite these weaknesses, $d_{\mathsf{FID}}$ currently remains the 'gold standard' for GAN evaluation and continuously attracts broad attention. In a recent study, Mathiasen & Hvilshøj (2020) proposed to compute eigenvalues rather than square root of a matrix as in $d_{\mathsf{FID}}$. We view this as a promising simplification and improvement, nonetheless a precise theoretical analysis has not been performed and therefore becomes the starting point of this paper.

The study of random matrix theory (RMT), with an emphasis on understanding the properties of (random) eigenvalues (Paul & Aue, 2014), has brought novel insights in the domain of deep learning (Liao & Couillet, 2018; Pastur, 2022; Baskerville et al., 2022), among which Seddik et al. (2020) analyzed deep learning representations of GAN generated images through the lens of eigenvalues of their sample covariance matrix (SCM). Driven by the need to efficiently quantify the distribution shift between two collections of heterogeneous data entries, we propose to compare sorted eigenvalues ($d_{\mathsf{Eig}}$) as a simple alternative to $d_{\mathsf{FID}}$. Our contributions are summarized as follows: For $i = 1, 2$, let $\mathbf{S}_i$ be the sample covariance of $\boldsymbol{Z}_i = (\boldsymbol{z}_1^i, \dots, \boldsymbol{z}_{n_i}^i)$ with $n_i$ $p$-dimensional vectors.

- ($d_{\mathsf{FID}}$) We articulate the fact that $\mathbf{S}_1\mathbf{S}_2$ is diagonalizable and has non-negative eigenvalues. This allows us to theoretically justify an improved algorithm of $d_{\mathsf{FID}}$, *i.e.*, by replacing the unique principal square root of a matrix with the element-wise square root of its eigenvalues. Therefore, the error bound for computing its eigenvalues is reduced to $\mathcal{O}(\varepsilon)\|\boldsymbol{S}_1\|\|\boldsymbol{S}_1\boldsymbol{S}_2\|$, reducing the run time by $\sim 25\%$.
- ($d_{\mathsf{Eig}}$) Since $\mathbf{S}_i$ is symmetric positive semidefinite, the error bound for computing its non-negative eigenvalues is further tightened to $\mathcal{O}(\varepsilon)\|\boldsymbol{S}_i\|$, along with reducing $\sim 90\%$ run time. From the viewpoint of random matrix theory (RMT), we demonstrate the asymptotically stable behavior of the largest eigenvalues (spikes).

## 2 THE IMPROVED $d_{\mathsf{FID}}$

*(Linear Algebra) Notation*: Lower case Roman or Greek letters (*e.g.*, $s, \epsilon, \gamma, \lambda$) denote scalars, bold lower case letters (*e.g.*, $\boldsymbol{v}, \boldsymbol{z}, \boldsymbol{\mu}$) denote vectors, and bold upper case letters (*e.g.*, $\boldsymbol{Q}, \boldsymbol{S}, \boldsymbol{U}, \boldsymbol{Z}, \boldsymbol{\Lambda}$) denote matrices. $^\mathsf{T}$ is matrix transpose, $\|.\|$ is $L^2$ norm, $\lesssim$ denotes asymptotically less than.

### 2.1 PRINCIPAL SQUARE ROOT OF A MATRIX

Without loss of accuracy, we discuss $d_{\mathsf{FID}}$ through the lens of linear algebra. More specifically, scalars, vectors and matrices discussed in the section are deterministic, while a statistical viewpoint on these objects will be later introduced in the proposed $d_{\mathsf{Eig}}$ section. For $i = 1, 2$, let $\boldsymbol{Z}_i = (\boldsymbol{z}_1^i, \dots, \boldsymbol{z}_{n_i}^i)$ be a collection of $n_i$ $p$-dimensional vectors. For simplicity, we assume sample mean $\frac{1}{n_i} \sum_{k=1}^{n_i} \boldsymbol{z}_k^i = \boldsymbol{0}$ throughout Sec. 2. Accordingly, $\boldsymbol{S}_i = \frac{1}{n_i} \boldsymbol{Z}_i \boldsymbol{Z}_i^\mathsf{T}$ denotes the sample covariance matrix (SCM) of $\boldsymbol{Z}_i$. We start the discussion with revisiting standard the definition(s) of $d_{\mathsf{FID}}$ (Givens & Shortt, 1984), then we elaborate the properties of principal square root – the key computational challenge of $d_{\mathsf{FID}}$.

#### 2.1.1 $\mathsf{Trace}((\mathbf{S}_1^{\frac{1}{2}}\mathbf{S}_2\mathbf{S}_1^{\frac{1}{2}})^{\frac{1}{2}})$

**Definition 1.** *Let $\boldsymbol{S}_i$ be the SCM of $\boldsymbol{Z}_i$ and w.l.o.g. $\boldsymbol{S}_1$ is non-singular, then we define*

$$d_{\mathsf{FID}}(\boldsymbol{S}_1, \boldsymbol{S}_2)^2 = \mathsf{Trace}(\boldsymbol{S}_1 + \boldsymbol{S}_2 - 2(\boldsymbol{S}_1^{\frac{1}{2}}\boldsymbol{S}_2\boldsymbol{S}_1^{\frac{1}{2}})^{\frac{1}{2}}). \qquad (1)$$

To compute the $\mathsf{Trace}()$ of $d_{\mathsf{FID}}$, we first need to clarify the symbol $\frac{1}{2}$ in Eq. 1. As mentioned in (Dowson & Landau, 1982), $\frac{1}{2}$ denotes the positive (or principal (Higham, 2008)) square root of a matrix $S$ such that $S^{\frac{1}{2}}S^{\frac{1}{2}} = S$, and 'principal' specifies the square root(s) $S^{\frac{1}{2}}$ with non-negative eigenvalues. In general, the square root of a matrix may neither exist nor be unique (Higham, 2008). Consider now the special case where $S$ is symmetric positive semidefinite (PSD), then we have

**Theorem 2.** *(**Principal square root**) Let a symmetric PSD $S$ be decomposed as $S = Q\Lambda Q^{\mathsf{T}}$, where $Q$ is an orthogonal matrix and $\Lambda$ is a diagonal matrix with non-negative eigenvalues, then $S^{\frac{1}{2}} := Q\Lambda^{\frac{1}{2}}Q^{\mathsf{T}}$ is the unique principal square root of $S$.*

Based on the definition of $S^{\frac{1}{2}}$, it is easy to see that $S^{\frac{1}{2}}S^{\frac{1}{2}} = S$. Importantly, $S^{\frac{1}{2}}$ is unique in the sense that if there exists another symmetric SPD $\tilde{S}^{\frac{1}{2}}$ such that $\tilde{S}^{\frac{1}{2}}\tilde{S}^{\frac{1}{2}} = S$, then $\tilde{S}^{\frac{1}{2}} = S^{\frac{1}{2}}$. With Thm. 2 in hand and given the fact that $S_1^{\frac{1}{2}}$ and $S_2$ are symmetric PSD, we make the following claim.

**Corollary 3.** $S_1^{\frac{1}{2}}S_2S_1^{\frac{1}{2}}$ *is a symmetric PSD matrix and therefore has a unique principal square root.*

**Claim**. Accordingly, $\mathsf{Trace}((S_1^{\frac{1}{2}}S_2S_1^{\frac{1}{2}})^{\frac{1}{2}})$ (Eq. 1) can be derived after eigenvalue decomposition suggested in Thm. 2. As a computational routine, this formulation is nevertheless undesirable. Because we need to call the eigenvalue decomposition function twice, which potentially increases computational time and error risk. Instead, we seek for another equivalent formulation of Eq. 1 (Givens & Shortt, 1984).

### 2.1.2 $\mathsf{Trace}((\mathbf{S}_1\mathbf{S}_2)^{\frac{1}{2}})$

**Lemma 4.** *Following the specifications of $S_i$ in Eq. 1, then we have*

$$d_{\mathsf{FID}}(S_1, S_2)^2 = \mathsf{Trace}(S_1 + S_2 - 2(S_1S_2)^{\frac{1}{2}}). \tag{2}$$

Because of non-singular $S_1$, it is not difficult to see that eigenvalues of $S_1^{\frac{1}{2}}S_2S_1^{\frac{1}{2}}$ and $S_1S_2$ are identical, and the corresponding eigenvectors are identical up to an invertible linear transformation $S_1^{\frac{1}{2}}$ (or $S_1^{-\frac{1}{2}}$). Due to the fact that $S_1^{\frac{1}{2}}S_2S_1^{\frac{1}{2}}$ is symmetric PSD, then we have

**Corollary 5.** $S_1S_2$ *is a diagonalizable matrix with non-negative eigenvalues and therefore has a unique principal square root.*

**Remark 6.** *Note that at least one of $S_1$ and $S_2$ should be non-singular, or the null space of $S_1$ should be contained in that of $S_2$ (Dowson & Landau, 1982). If $S_1$ is singular, then the above discussions remain the same after switching the role of $S_1$ and non-singular $S_2$.*

**Claim**. Consequently, eigenvalues of $(S_1S_2)^{\frac{1}{2}}$ are mathematically equivalent to the element-wise square root of eigenvalues of $S_1S_2$. Since $S_1S_2$ and $S_1^{\frac{1}{2}}S_2S_1^{\frac{1}{2}}$ have identical eigenvalues and the trace of a diagonalizable matrix is the sum over its eigenvalues, then we have $\mathsf{Trace}((S_1S_2)^{\frac{1}{2}}) = \mathsf{Trace}((S_1^{\frac{1}{2}}S_2S_1^{\frac{1}{2}})^{\frac{1}{2}})$ and prove Lem. 4. Importantly, this rigorously justifies the workaround algorithm of $d_{\mathsf{FID}}$ that computes the element-wise square root of eigenvalues, which bypasses the expansive computation of the square root of a matrix.

### 2.2 Element-wise square root of eigenvalues

Before substituting the square root component of $d_{\mathsf{FID}}$, let us take a step back and re-examine its widely-used implementation scipy.lingalg.sqrtm()[1]. In a nutshell, the underlying computational routine is a blocked Schur algorithm (Björck & Hammarling, 1983; Deadman et al., 2012), which includes two phases: Schur decomposition (schur()) and solving (triangular) Sylvester equation. For computing $\mathsf{Trace}()$ of $p \times p$ diagonalizable matrix $(S_1S_2)^{\frac{1}{2}}$, we show the latter phase is redundant.

---

[1]https://github.com/GaParmar/clean-fid/blob/main/cleanfid/fid.py

### 2.2.1 NUMERICAL ERROR BOUND

**Corollary 7.** *(Schur decomposition) Let the diagonalizable matrix $S_1 S_2$ be decomposed as $QUQ^\mathsf{T}$. Here, $Q$ is an orthogonal matrix and $U$ is an upper triangular matrix. Then* $\mathsf{Trace}((S_1 S_2)^{\frac{1}{2}}) = \sum_{j=1}^{p} \sqrt{u_{jj}}$, *where $u_{11}, \ldots, u_{pp}$ are diagonal entries of $U$.*

This equation is derived from the fact that diagonal entries of $U$ are exactly (non-negative) eigenvalues of $S_1 S_2$. As an immediate consequence, it suffices to compute schur() for obtaining $\mathsf{Trace}((S_1 S_2)^{\frac{1}{2}})$. By default, schur() simultaneously computes $U$ and $Q$. In our case, we only want to compute diagonal entries of $U$. This leads to a more speedy eigvals() that shares the same core routine as schur(). Eventually, we replace the standard pythonic implementation np.trace(sqrtm()) with np.sqrt(eigvals()).sum() (See Fig. 1 for more details and Mathiasen & Hvilshøj (2020) for reference). Such a series of algorithmic simplification allows us to propose a (strictly) tighter error bound compared to the original case *w.r.t.* sqrtm().

**Error bound of eigvals().** As discussed in (Anderson et al., 1999), for the computed eigenvalue $\hat{\gamma}_j$ and eigenvalue $\gamma_j$ of $S_1 S_2$ we have $|\hat{\gamma}_j - \gamma_j| \lesssim \mathcal{O}(\epsilon) s_j^{-1} \|S_1 S_2\|$, where $\epsilon$ is machine epsilon. The remaining task is to compute $s_j$. Since if $v_j$ is the right eigenvector for $\gamma_j$, then the left eigenvector is $v_j^\mathsf{T} S_1^{-1}$. Because of $s_j := |v_j^\mathsf{T} S_1^{-1} v_j| = \|S_1^{-\frac{1}{2}} v_j\|^2$ we have $s_j^{-1} \leq \|S_1^{\frac{1}{2}}\|^2 = \|S_1\|$. For $j = 1, \ldots, p$, the (asymptotic) error bound for computing eigenvalue $\hat{\gamma}_j$ can be formulated as

$$|\hat{\gamma}_j - \gamma_j| \lesssim \mathcal{O}(\epsilon) \|S_1\| \|S_1 S_2\|. \tag{3}$$

Moreover, if we want to compute eigenvalues of the $p \times p$ SCM $S_i$ that is symmetric PSD, we can utilize the eigvalsh() with lower run time and obtain a tighter error bound.

**Error bound of eigvalsh().** For $j = 1, \ldots, p$, the error bound for computing eigenvalue $\lambda_j^i$ of $S_i$ can be formulated as (Anderson et al., 1999)

$$|\hat{\lambda}_j^i - \lambda_j^i| \leq \mathcal{O}(\epsilon) \|S_i\|. \tag{4}$$

### 2.2.2 EIGENVALUE COMPARISON

Now, we discuss an important variant of $d_{\mathsf{FID}}$ when $S_1$ and $S_2$ commute, *i.e.*, $S_1 S_2 = S_2 S_1$.

**Corollary 8.** *(Unsorted eigenvalue comparison) Let $S_{1,2}$ be two SCMs that are simultaneously diagonalizable by an orthogonal matrix $Q$, then*

$$d_{\mathsf{FID}}(S_1, S_2)^2 = \mathsf{Trace}((S_1^{\frac{1}{2}} - S_2^{\frac{1}{2}})^2) = \sum_{j=1}^{p} (\sqrt{\tilde{\lambda}_j^1} - \sqrt{\tilde{\lambda}_j^2})^2, \tag{5}$$

*where $\tilde{\lambda}_j^i$ is the $j$-th eigenvalue of $S_i$ w.r.t. $Q$.*

Under such a special case where $S_1$ and $S_2$ share the same eigenbasis, $d_{\mathsf{FID}}$ is reduced to computing the Euclidean distance between unsorted eigenvalues. Motivated by this reduction, we propose to compare sorted eigenvalues as a simple alternative to $d_{\mathsf{FID}}$.

**Definition 9.** *(Sorted eigenvalue comparison) Let $S_{1,2}$ be two SCMs, then we define*

$$d_{\mathsf{Eig}}(S_1, S_2)^2 = \sum_{j=1}^{p} (\sqrt{\lambda_j^1} - \sqrt{\lambda_j^2})^2, \tag{6}$$

*where $\lambda_j^i$ is the $j$-th **largest** eigenvalue of $S_i$. Accordingly, $d_{\mathsf{Eig}}$ is a pseudometric on the set of SCMs with order $p$.*

Note that $S_1$ and $S_2$ in Eq. 6 do not necessarily commute. Instead of eigvals() used for computing eigenvalues of non-symmetric $S_1 S_2$ (Eq. 2), $d_{\mathsf{Eig}}$ can be obtained with a more numerically stable and faster eigvalsh(), which is customized to compute $\lambda_j^i$ of symmetric $S_i$. As a pseudometric, $d_{\mathsf{Eig}}$ satisfies non-negativity, symmetry and triangular inequality, while SCMs need not to be indistinguishable regarding $d_{\mathsf{Eig}}$. Following the convention, $d_{\mathsf{Eig}}$ and $d_{\mathsf{FID}}$ scores reported in the following are always the square of $d_{\mathsf{Eig}}$ and $d_{\mathsf{FID}}$ resp.

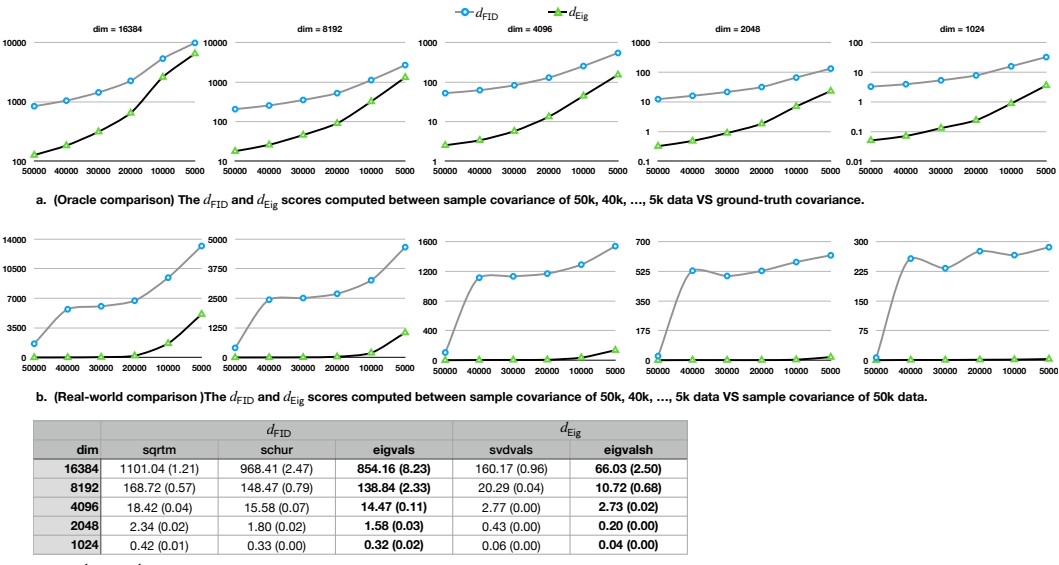

a. (Oracle comparison) The $d_{\mathsf{FID}}$ and $d_{\mathsf{Eig}}$ scores computed between sample covariance of 50k, 40k, ..., 5k data VS ground-truth covariance.

b. (Real-world comparison )The $d_{\mathsf{FID}}$ and $d_{\mathsf{Eig}}$ scores computed between sample covariance of 50k, 40k, ..., 5k data VS sample covariance of 50k data.

| dim | $d_{\mathsf{FID}}$ | | | $d_{\mathsf{Eig}}$ | |
|---|---|---|---|---|---|
| | sqrtm | schur | eigvals | svdvals | eigvalsh |
| 16384 | 1101.04 (1.21) | 968.41 (2.47) | **854.16 (8.23)** | 160.17 (0.96) | 66.03 (2.50) |
| 8192 | 168.72 (0.57) | 148.47 (0.79) | **138.84 (2.33)** | 20.29 (0.04) | 10.72 (0.68) |
| 4096 | 18.42 (0.04) | 15.58 (0.07) | **14.47 (0.11)** | 2.77 (0.00) | 2.73 (0.02) |
| 2048 | 2.34 (0.02) | 1.80 (0.02) | **1.58 (0.03)** | 0.43 (0.00) | 0.20 (0.00) |
| 1024 | 0.42 (0.01) | 0.33 (0.00) | **0.32 (0.02)** | 0.06 (0.00) | 0.04 (0.00) |

c. The $d_{\mathsf{FID}}$ and $d_{\mathsf{Eig}}$ run time obtained with different implementation variants.

Figure 2: **The toy studies of multivariate Gaussian distribution** $(\mathbf{0}, \boldsymbol{\Sigma})$. Here, all the experiments are computed with Intel(R) i9-9940X CPU @ 3.30GHZ and repeated with four random seeds. Since the coefficient of variance std/mean $< 0.01$ for both $d_{\mathsf{Eig}}$ and $d_{\mathsf{FID}}$, we only report the mean score in Plot a and b. Besides, sqrtm(), schur() and eigvals() achieve identical numerical results for computing $d_{\mathsf{FID}}$ up to negligible rounding error. So do svdvals() and eigvalsh() for computing $d_{\mathsf{Eig}}$. Therefore, only two curves are presented in Plot a, b.

### 2.2.3 TOY STUDIES: $d_{\mathsf{Eig}}$ IS MORE RELIABLE THAN $d_{\mathsf{FID}}$.

**Mathematical equivalence between $d_{\mathsf{Eig}}$ and $d_{\mathsf{FID}}$.** For proof of principle, we conduct toy studies with multivariate Gaussian data. Concretely, we construct non-negative diagonal entries of a $p$-dim covariance matrix $\boldsymbol{\Sigma}$ with np.abs(np.random.randn($p$)), while keeping the off-diagonal entries zero. By multiplying $\boldsymbol{\Sigma}^{\frac{1}{2}}$ and $\boldsymbol{X}_i = $ np.random.randn($p, n_i$), we obtain $n_i$ Gaussian data entries $\boldsymbol{Y}_i = \boldsymbol{\Sigma}^{\frac{1}{2}} \boldsymbol{X}_i$ that are drawn from $(\mathbf{0}, \boldsymbol{\Sigma})$. Then we compare $\boldsymbol{S}_1 = \frac{1}{n_1} \boldsymbol{Y}_1 \boldsymbol{Y}_1^{\mathsf{T}}$ to ground-truth $\boldsymbol{\Sigma}$ (Fig. 2(a)) and to $\boldsymbol{S}_2 = \frac{1}{n_2} \boldsymbol{Y}_2 \boldsymbol{Y}_2^{\mathsf{T}}$ (Fig. 2(b)). Following above theoretical discussions, we instantiate Eq. 6 of $d_{\mathsf{Eig}}$ with sqrtm(), schur() and eigvals(). Because $\boldsymbol{S}_i$ is a symmetric SPD, we implement Eq. 2 of $d_{\mathsf{FID}}$ with svdvals() and eigvalsh(). Throughout our experiments, we notice that the results of implementation variants are identical up to very small rounding errors. Therefore, we experimentally confirm the validity of improved $d_{\mathsf{FID}}$ and the equivalency between svdvals() and eigvalsh(). Because identical (sample) covariances are simultaneously diagonalizable, we have $d_{\mathsf{Eig}} = d_{\mathsf{FID}}$ in theory. Since $\boldsymbol{S}_1 \approx \boldsymbol{S}_2 \approx \boldsymbol{\Sigma}$ with sufficient amount of data, we expect $d_{\mathsf{Eig}} \approx d_{\mathsf{FID}} \approx 0$ in practice.

**Numerical difference between $d_{\mathsf{Eig}}$ and $d_{\mathsf{FID}}$.** When comparing $\boldsymbol{S}_1$ to $\boldsymbol{\Sigma}$, Fig. 2 (a) shows that $d_{\mathsf{Eig}}$ and $d_{\mathsf{FID}}$ have a comparable trend of decreasing scores with a growing number of data entries ($5k \rightarrow 50k$). This indicates that both $d_{\mathsf{Eig}}$ and $d_{\mathsf{FID}}$ are meaningful metrics and can converge to their theoretical limit. When comparing $\boldsymbol{S}_1$ to $\boldsymbol{S}_2$, Fig. 2 (b) illustrates that $d_{\mathsf{Eig}}$ is more resistant to the data size difference. In contrast to $d_{\mathsf{FID}}$, it suffices to use a smaller amount of data to achieve a good estimation for $d_{\mathsf{Eig}}$. Arguably, $d_{\mathsf{Eig}}$ represents a more reliable score than $d_{\mathsf{FID}}$ due to the fact that 1) $d_{\mathsf{Eig}}$ demonstrates favorable convergence curves that are overall closer to 0, and 2) in comparison with the standard $d_{\mathsf{FID}}$ (Eq. 2), $d_{\mathsf{Eig}}$ (Eq. 6) is a more faithful routine to approximate Eq. 5 – the simplified $d_{\mathsf{FID}}$ for our toy setting.

**Run time**. When comparing different variants for implementing $d_{\mathsf{Eig}}$ and $d_{\mathsf{FID}}$, Fig. 2 (c) shows $18\% - 32\%$ reduction of run time by replacing sqrtm() with eigvals(), and we further reduce the run time by $85\% - 94\%$ when utilizing eigvalsh(). As a result, it is beneficial to apply the improved $d_{\mathsf{Eig}}$ and proposed $d_{\mathsf{FID}}$ for computing distribution shifts, especially in the high dimensional cases such as $p = 16384$. From now on, $d_{\mathsf{Eig}}$ and $d_{\mathsf{FID}}$ are computed with eigvals() and eigvalsh() by default.

## 3 THE PROPOSED $d_{\mathsf{Eig}}$

*(Probability) Notation*: Sans serif lower case letters (*e.g.*, x, y, z, $\lambda$) denote random variables, sans serif bold lower case letters (*e.g.*, **x**, **y**, **z**, $\boldsymbol{\mu}$) denote random vectors, and sans serif bold upper case letters (*e.g.*, **X**, **Y**, **Z**, **Q**, **S**) denote random matrices. $\asymp$ denotes asymptotic equivalence, $\mathbb{N}_+$ is the set of positive integers, $\mathbb{E}$ is expectation and $\mathbb{P}$ is probability distribution.

Following the computational analysis on deterministic eigenvalues, we here investigate the statistical facets of $d_{\mathsf{Eig}}$ from the viewpoint of RMT. Given a SCM, researchers have a keen interest in understanding the asymptotic behavior of the largest eigenvalues (spikes) (Izenman, 2021). This is motivated by the observation that spikes reflect the direction of largest variance and reserve the most critical information (Perry et al., 2018). Since spikes also dominate the computation of $d_{\mathsf{Eig}}$, it is important to analyze their asymptotic behavior in our study.

### 3.1 ASYMPTOTICALLY STABLE BEHAVIOR

In this section, eigenvalue, vector and matrix are non-deterministic and indicate random eigenvalue (variable), random vector and random matrix unless stated otherwise. Firstly, we recall a canonical case where $\mathbf{X} = (\mathbf{x}_1, \ldots, \mathbf{x}_n)$ is a random matrix with IID entries. For $j = 1, \ldots, p$ and $k = 1, \ldots, n$, let $n \asymp p$ and $x_{jk}$ be the IID entries of $\mathbf{X}$ satisfying $\mathbb{E}|x_{jk}|^2 = 1$ and $\mathbb{E}|x_{jk}|^m \leq c_m$ for all fixed $m \in \mathbb{N}_+$. Similar to Sec. 2, we further assume $\mathbb{E}x_{jk} = 0$. As shown in one of the pioneer studies (Marčenko & Pastur, 1967), the asymptotic eigenvalue density of $\frac{1}{n}\mathbf{X}\mathbf{X}^{\mathsf{T}}$ can be well characterized with limiting Stieltjes transform. However, the assumptions of IID entries and diagonal covariance structure are very stringent and do not reflect real-world data statistics. Regarding GAN assessment as a concrete example, samples drawn from $\mathbf{x}_j$ are usually representations obtained with the penultimate layer (*pool3*) of an Inception V3 model (Szegedy et al., 2016). Therefore, $x_{1k}, \ldots x_{pk}$ of $\mathbf{x}_k$ can be dependent and have a more general covariance structure $\boldsymbol{\Sigma}$. To resolve the gap, we assume $\boldsymbol{\Sigma}$ satisfies the stability condition as in (Bao et al., 2015, Condition 1.1 (iii)) and propose to investigate $\mathbf{Y} = \boldsymbol{\Sigma}^{\frac{1}{2}}\mathbf{X}$, a linear transformation of $\mathbf{X}$.

**Theorem 10.** *(Case of zero expectation:* $\mathbf{Y} = \boldsymbol{\Sigma}^{\frac{1}{2}}\mathbf{X}$*) Fix $r$ and let* $\bar{\lambda}_1 \geq \cdots \geq \bar{\lambda}_r$ *be the $r$ largest eigenvalues of* $\mathbf{Q} = \frac{1}{n}\mathbf{Y}\mathbf{Y}^{\mathsf{T}}$*, then for any $j = 1, \ldots, r$ we can find deterministic $\bar{\theta}_j$ such that for any (small) $\alpha > 0$ and (big) $\beta > 0$, we have*

$$\mathbb{P}(|\bar{\lambda}_j - \bar{\theta}_j| \geq n^{-\frac{2}{3}+\alpha}) \leq c_{\alpha,\beta}n^{-\beta} \tag{7}$$

*for some constant $c_{\alpha,\beta}$ independent of $n, p$.*

**Discussion**. Here, Thm. 10 is a direct result of the local density law (Knowles & Yin, 2017). Note that $\frac{1}{n}\mathbb{E}\mathbf{Y}\mathbf{Y}^{\mathsf{T}} = \boldsymbol{\Sigma}$ holds true and we impose linear dependency among $x_{1k}, \ldots x_{pk}$ of $\mathbf{x}_k$ to approximate real-world scenario. In the meantime, $\mathbf{x}_1, \ldots, \mathbf{x}_n$ remain identically distributed, which reflects the key fact that data entries such as image representations of a GAN model are drawn from the same probability distribution. As learned representations commonly have non-zero expectations $\boldsymbol{m} \neq \mathbf{0}$, we further introduce a deterministic rank-1 matrix $\boldsymbol{M} = \boldsymbol{m}\boldsymbol{e}^{\mathsf{T}}$ to model this scenario. Here, $\boldsymbol{e} = (1, \ldots, 1)^{\mathsf{T}}$ and $\boldsymbol{m} = d\boldsymbol{v}$ satisfying $d \asymp p \asymp n$ and $\|\boldsymbol{v}\| = 1$. For $\mathbf{Z} = \boldsymbol{M} + \boldsymbol{\Sigma}^{\frac{1}{2}}\mathbf{X}$, we have

**Lemma 11.** *(Case of non-zero expectation:* $\mathbf{Z} = \boldsymbol{M} + \boldsymbol{\Sigma}^{\frac{1}{2}}\mathbf{X}$*) Fix $r$ and let $\lambda_1 \geq \cdots \geq \lambda_r$ be the $r$ largest eigenvalues of* $\mathbf{S} = \frac{1}{n}\mathbf{Z}\mathbf{Z}^{\mathsf{T}}$*, then for any $j = 1, \ldots, r$ we can find deterministic $\theta_j$ such that for any (small) $\alpha > 0$ and (big) $\beta > 0$, we have*

$$\mathbb{P}(|\lambda_j - \theta_j| \geq n^{-\frac{1}{2}+\alpha}) \leq c_{\alpha,\beta}n^{-\beta} \tag{8}$$

*for some constant $c_{\alpha,\beta}$ independent of $n, p$.*

**Discussion**. As shown in (Bai, 1999, Lemma 2.2), the eigenvalue counting functions of $\mathbf{Q}$ and $\mathbf{S}$ differ by at most $\frac{1}{p}$. Thus, $\bar{\lambda}_{j+1} \leq \lambda_j \leq \bar{\lambda}_{j-1}$ for $j = 2, 3, ..., r$. Then, the rigidity estimation of $\lambda_j$ for $j = 2, 3, ..., r$ can be obtained by considering $\theta_j := \bar{\theta}_j$ and applying Thm. 10. As to the case of $j = 1$, we consider Eq. 8 for $\lambda_1(\geq \bar{\lambda}_1)$ as a conjecture and leave the proof for future work. Together with discussions in Sec. 2, we illustrate both the numerical and asymptotic stability of $d_{\mathsf{Eig}}$.

**Remark 12.** *Similar to (Louart & Couillet, 2018, Remark 0.1), the above lemma suggests a (abusive) definition of SCM* $\mathbf{S} = \frac{1}{n}\mathbf{Z}\mathbf{Z}^{\mathsf{T}}$ *without subtracting the mean expectation. Taking $\mathbf{S}$ as the input of $d_{\mathsf{Eig}}$, we show that it is feasible for $d_{\mathsf{Eig}}$ to quantify the distribution shift in follow-up GAN studies.*

### 3.2 GAN STUDIES: $d_{\mathsf{Eig}}$ IS A SIMPLE ALTERNATIVE TO $d_{\mathsf{FID}}$.

Recently, Parmar et al. (2022) discovered surprising subtleties of image pre-processing steps for downstream GAN evaluation. To faithfully benchmark the GAN performance of state-of-the-art (sota) models, the authors published new APIs to reproduce the evaluation results. Hence, the implementation of our GAN experiments is built on top of these APIs. Next, we summarize four key aspects of GAN evaluation that we examine in this study.

**4 Scores**. Similar to Parmar et al. (2022), we take two widely-used scores $d_{\mathsf{FID}}$ and $d_{\mathsf{KID}}$ as baselines. Then, we investigate two variants of the proposed metric: $d_{\mathsf{Eig}}(\boldsymbol{S}_1, \boldsymbol{S}_2)^2 = \sum_{j=1}^{p}(\sqrt{\lambda_j^1} - \sqrt{\lambda_j^2})^2$ (Eq. 6) and $d'_{\mathsf{Eig}}(\boldsymbol{S}_1, \boldsymbol{S}_2)^2 = \sum_{j=1}^{p}(\sqrt{\bar{\lambda}_j^1} - \sqrt{\bar{\lambda}_j^2})^2 + \|\boldsymbol{m}_1 - \boldsymbol{m}_2\|^2$. The $\lambda_j^i$ of the former are eigenvalues of $\boldsymbol{S}_i$, and $\bar{\lambda}_j^i$ of the latter are eigenvalues of $\boldsymbol{S}_i - \frac{1}{n_i}\boldsymbol{m}_i\boldsymbol{m}_i^{\mathsf{T}}$, where $\boldsymbol{m}_i$ is the sample mean. Differing from toy settings of Gaussian distribution $(\boldsymbol{0}, \boldsymbol{\Sigma})$ that lead to $d_{\mathsf{Eig}} \approx d_{\mathsf{FID}} \approx 0$ with sufficient data, we do not have such a theoretical limit or ground-truth score in GAN studies. As a workaround, we consider $d_{\mathsf{FID}}$ to be the 'gold standard' score for analyzing $d_{\mathsf{Eig}}$. Without loss of accuracy, we take $d_{\mathsf{KID}} \times 10^3$ and $d_{\mathsf{Eig}} \times 10$ for clearer comparisons.

**3 Models**. To illustrate the strength of $d_{\mathsf{Eig}}$ for challenging cases, we investigate three sota GAN models and probe their nuances when visual evaluations are non-trivial: StyleGAN2 with the recommended Config (**Style2**) (Karras et al., 2020a), StyleGAN3 with translation equivariance Config (**Style3t**) and with translation and rotation equivariance Config (**Style3r**) (Karras et al., 2021).

**3 Interpolations**. Following the practice of Parmar et al. (2022), we also present results that are influenced by different image interpolations such as Clean (**Clean**), PyTorch_legacy (**Py_legacy**) and TensorFlow_legacy (**TF_legacy**).

**5 Datasets**. Lastly, we run thorough comparisons on commonly-used datasets including FFHQ, AFHQ, and LSUN (Horse, Church, Cat categories) for GAN model training. For each dataset, we generate 100k fake images and repeat each experiment 4 times by randomly sampling a given number of image entries from 100k fake images.

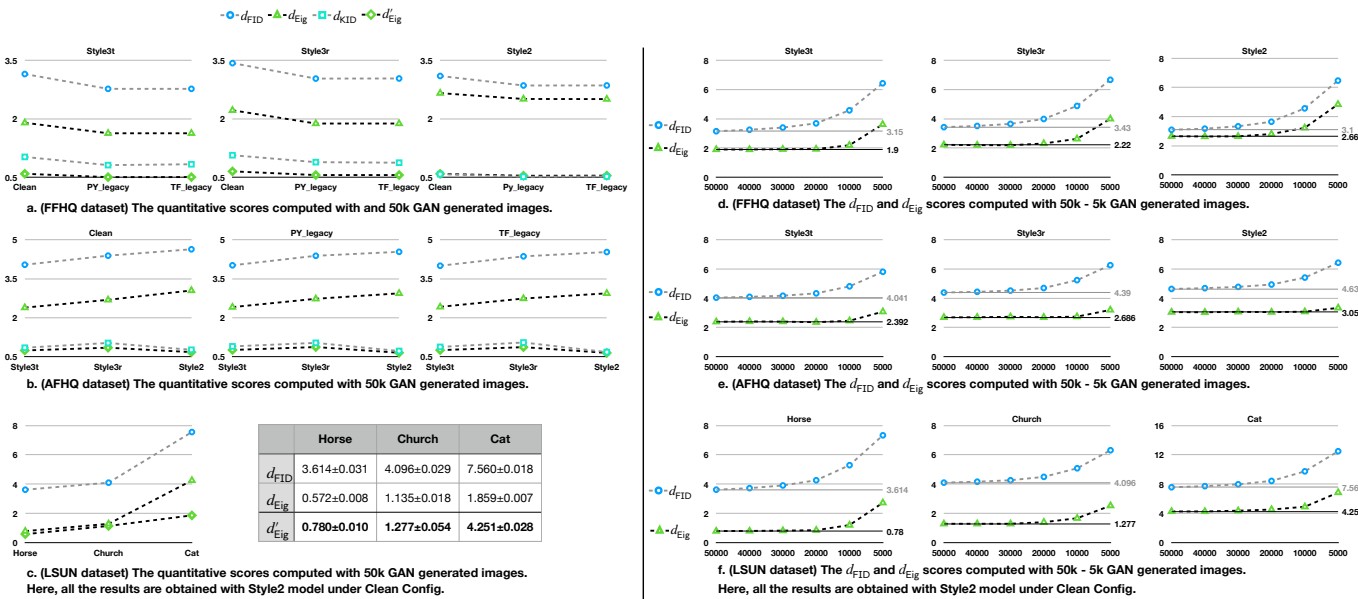

**Figure 3: The main results of GAN studies**. Here, 70k, 15803 and 50k real images for FFHQ, AFHQ and LSUN datasets resp. are applied to compute the reported scores.

In the following, we discuss the main results of our GAN studies. As displayed in Fig. 3 (a, b), $d_{\mathsf{Eig}}$ and $d_{\mathsf{FID}}$ show similar evaluation curves and correlate well with each other in terms of different combinations of models and interpolations. When observing the convergence curve with an increasing

amount of GAN generated images Fig. 3 (d, f), we observe an identical behavior as in the toy studies. That is, $d_{\mathsf{Eig}}$ is more favorable than $d_{\mathsf{FID}}$ in the sense that it suffices to use a small amount of image entries to obtain a good estimation for $d_{\mathsf{Eig}}$. Similar claims can be made for the LSUN dataset. As shown in Fig. 3 (c), $d_{\mathsf{Eig}}$ illustrates comparably increasing scores from the Horse to the Cat category, indicating less satisfying GAN generation results for the Cat images. Meanwhile, the convergence speed remains faster for $d_{\mathsf{Eig}}$ compared to $d_{\mathsf{FID}}$ (Fig. 3 (f)). With regard to $d'_{\mathsf{Eig}}$, the variant of sorted eigenvalue comparison show less consistency with the gold standard $d_{\mathsf{FID}}$ (See Fig. 3 (a, b)) and is less desirable in our GAN studies. Based on the investigations of the four key aspects and theoretical advantages of $d_{\mathsf{Eig}}$ discussed above, the proposed $d_{\mathsf{Eig}}$ represents a simple alternative to $d_{\mathsf{FID}}$. By applying $d_{\mathsf{Eig}}$ in GAN model evaluation, we take a critical step towards a more comprehensive analysis of high-dim distribution shift between two collections of image entries.

| FFHQ | Eigenvalue | Eigenvector |
|---|---|---|
| 1 | 259.14±0.157 | 0.99 |
| 2 | 15.92±0.048 | 0.002 |
| 3 | 10.24±0.047 | 0.009 |
| 4 | 7.31±0.024 | 0.001 |
| 5 | 6.85±0.049 | 0.013 |
| 6 | 5.41±0.019 | 0.004 |
| 7 | 3.73±0.023 | 0.003 |
| 8 | 3.59±0.009 | 0.002 |
| 9 | 3.11±0.007 | 0.001 |
| 10 | 2.72±0.001 | 0.004 |

| AFHQ | Eigenvalue | Eigenvector |
|---|---|---|
| 1 | 133.53±0.201 | 0.99 |
| 2 | 14.43±0.037 | 0.015 |
| 3 | 9.18±0.034 | 0.012 |
| 4 | 6.43±0.070 | 0.009 |
| 5 | 5.5±0.039 | 0.003 |
| 6 | 4.25±0.016 | 0.003 |
| 7 | 3.28±0.007 | 0.005 |
| 8 | 2.82±0.009 | 0.0003 |
| 9 | 2.44±0.004 | 0.002 |
| 10 | 2.3±0.015 | 0.002 |

| Horse | Eigenvalue | Eigenvector |
|---|---|---|
| 1 | 237.81±0.197 | 0.99 |
| 2 | 16.46±0.025 | 0.011 |
| 3 | 6.06±0.012 | 0.002 |
| 4 | 4.14±0.013 | 0.006 |
| 5 | 4.04±0.015 | 0.003 |
| 6 | 3.10±0.013 | 0.0003 |
| 7 | 2.70±0.015 | 0.016 |
| 8 | 2.56±0.009 | 0.002 |
| 9 | 2.41±0.010 | 0.013 |
| 10 | 2.12±0.014 | 0.001 |

| Church | Eigenvalue | Eigenvector |
|---|---|---|
| 1 | 186.59±0.212 | 0.99 |
| 2 | 8.81±0.038 | 0.005 |
| 3 | 7.33±0.015 | 0.004 |
| 4 | 4.24±0.011 | 0.012 |
| 5 | 3.92±0.015 | 0.015 |
| 6 | 3.21±0.009 | 0.014 |
| 7 | 2.53±0.012 | 0.023 |
| 8 | 2.28±0.011 | 0.005 |
| 9 | 1.92±0.005 | 0.026 |
| 10 | 1.88±0.007 | 0.009 |

| Cat | Eigenvalue | Eigenvector |
|---|---|---|
| 1 | 291.90±0.094 | 0.99 |
| 2 | 11.21±0.025 | 0.012 |
| 3 | 6.27±0.022 | 0.010 |
| 4 | 4.39±0.014 | 0.002 |
| 5 | 4.12±0.009 | 0.001 |
| 6 | 3.29±0.013 | 0.007* |
| 7 | 2.79±0.017 | 0.0005* |
| 8 | 2.63±0.016 | 0.015 |
| 9 | 2.44±0.005 | 0.012* |
| 10 | 2.15±0.004 | 0.016 |

Figure 4: **The eigenvalue fluctuation and eigenvector similarity for 10 largest spikes of** $d_{\mathsf{Eig}}$. Here, all the experiments are obtained with Style2 under Clean configuration. The eigenvalue fluctuation (standard deviation) is obtained by repeating experiments with 4 random seeds. Also, we report the largest cosine similarity between the $i$-th largest eigenvector of GAN images and its counterpart of real images. The * indicates that the largest cosine similarity is not obtained between the the i-th largest eigenvectors of GAN and real images.

As displayed in Fig. 4, we report the eigenvalue and eigenvector behaviors for the 10 largest spikes. The reported cutoffs were determined by the dominant percentage ($> 80\%$) taken by these spikes compared to the complete spectrum. Notably, the 10 largest spikes present small fluctuations (std) obtained with four random seeds, which serves as complementary evidence to support the theoretical rigidity estimations discussed in Thm. 10 and Lem. 11. Except for the few cases marked with *, the largest cosine similarity is mostly obtained with the $i$-th largest eigenvector for both GAN and real images. If we decompose the distribution shift to scale shift (eigenvalue shift) and rotation shift (eigenvector shift), such results suggest that the dominant eigenvector shift is only determined by the cosine of the angle between them, and is not influenced by eigenvector permutation. By weighing the estimation challenges of eigenvectors, $d_{\mathsf{Eig}}$ makes a meaningful trade-off that only takes eigenvalue differences into account.

| | Clean | | | PY_legacy | | | TF_legacy | | |
|---|---|---|---|---|---|---|---|---|---|
| | Style3t | Style3r | Style2 | Style3t | Style3r | Style2 | Style3t | Style3r | Style2 |
| $d_{\mathsf{FID}}$ | 3.150±0.027 | 3.43±0.042 | 3.1±0.01 | 2.77±0.027 | 3.033±0.041 | 2.856±0.016 | 2.77±0.022 | 3.036±0.037 | 2.857±0.013 |
| $d_{\mathsf{Eig}}$ | 1.9±0.06 | 2.22±0.061 | 2.657±0.073 | 1.627±0.060 | 1.883±0.053 | 2.513±0.056 | 1.625±0.005 | 1.883±0.043 | 2.507±0.049 |

Figure 5: **The nuance between** $d_{\mathsf{FID}}$ **and** $d_{\mathsf{Eig}}$. Here, all the experiments are conducted with the FFHQ dataset.

Lastly, we report a nuanced case when comparing $d_{\mathsf{FID}}$ and $d_{\mathsf{Eig}}$. Fig. 5 shows that the face generalization performance *w.r.t.* $d_{\mathsf{Eig}}$ tends to be improved from Style2 to Style3r and Style3t, which is not compatible with $d_{\mathsf{FID}}$. By imposing the translation and rotation equivariance in StyleGAN3, Karras et al. (2021) reported anti-aliasing improvements over StyleGAN2 by resolving the 'texture sticking' issue. Such clear visual improvements are supported by the decreasing $d_{\mathsf{Eig}}$ scores. However, due to the lack of ground-truth, whether such a correlation between visual improvements and $d_{\mathsf{Eig}}$ supports the effectiveness of $d_{\mathsf{Eig}}$ remains inconclusive.

## 4    RELATED WORK

**Fast $d_{\mathsf{FID}}$.** The role of eigenvalues played in $d_{\mathsf{FID}}$ has been firstly noticed in the study of Fast Fréchet Inception Distance (Mathiasen & Hvilshøj, 2020). When $n_1 \ll p$, the researchers suggested to compute the eigenvalues of $n_1 \times n_1$ matrix $\boldsymbol{Z}_1^\mathsf{T} \boldsymbol{Z}_2 \boldsymbol{Z}_2^\mathsf{T} \boldsymbol{Z}_1$ to supervise the model training. Accordingly, the differences between our study and Fast $d_{\mathsf{FID}}$ lie in the fact that we do not assume $n_1 \ll p$ and we compute $\boldsymbol{S}_1 \boldsymbol{S}_2 = \frac{1}{n_1 n_2} \boldsymbol{Z}_1 \boldsymbol{Z}_1^\mathsf{T} \boldsymbol{Z}_2 \boldsymbol{Z}_2^\mathsf{T}$ instead. Since a more precise justification *w.r.t.* eigenvalues was not present in Mathiasen & Hvilshøj (2020), the key contributions of our theoretical analysis on $d_{\mathsf{FID}}$ come from articulating the unique principal square root, diagonalizable $\boldsymbol{S}_1 \boldsymbol{S}_2$ with non-negative eigenvalues and proposing a tight asymptotic error bound.

**Seddik et al. (2020).** From a RMT viewpoint, Seddik et al. (2020) studied the SCM of GAN image representations and argued that such representations behave asymptotically as if they are drawn from a Gaussian mixture. Following this insight, we further show that comparing sorted eigenvalues of SCMs is useful and efficient for measuring high-dimensional distribution shift, which is a novel and distinct contribution by our theoretical study.

## 5    LIMITATIONS AND FUTURE WORK

### 5.1    EIGENVECTOR

In contrast to the improved $d_{\mathsf{FID}}$ that implicitly takes eigenvectors of $\boldsymbol{S}_i$ into account via the matrix multiplication $\boldsymbol{S}_1 \boldsymbol{S}_2$, the proposed $d_{\mathsf{Eig}}$ only measures the eigenvalue difference. Admittedly, the exclusion of eigenvectors in $d_{\mathsf{Eig}}$ is mainly due to the disencouraging properties such as more loose numerical error bound (Anderson et al., 1999) and more strict conditions for distribution estimation (Knowles & Yin, 2013). Nevertheless, eigenvectors carry plausibly critical information and should be carefully examined in subsequent work.

### 5.2    FUTURE STUDIES: $d_{\mathsf{Eig}}$ MAY BE MORE COMPREHENSIVE AND INFORMATIVE THAN $d_{\mathsf{FID}}$.

Similar to existing measurements, $d_{\mathsf{Eig}}$ remains a scalar-valued score for measuring high-dimensional distribution shifts. A more comprehensive quantification is still missing for applications in both the natural and medical image domains. Due to the inherent data heterogeneity and critical implications for real-world application, facilitating in-depth analysis of distribution shifts underlying high-dimensional images (or representations) is of key importance to support the development and application of high-quality data science approaches, *e.g.*, in the medical domain (Yue et al., 2020; Cios & Moore, 2002). In such a scenario where inaccurate analysis can have severe consequences, existing scalar-valued scores including $d_{\mathsf{Eig}}$ is not sufficient. To resolve this issue, a direct follow-up on $d_{\mathsf{Eig}}$ is to individually compare the eigenvalue difference along each dimension. Naturally, the scalar-valued $d_{\mathsf{Eig}}$ is decomposed to a multi-dimensional vector-valued measurement and enables a more complete overview of data heterogeneity. In addition, the $d_{\mathsf{Eig}}$ builds the bridge between the classical principal component analysis (PCA) (Abdi & Williams, 2010) and latent semantic understanding (Shen & Zhou, 2021; Härkönen et al., 2020).

Taking cancer studies as an example(Fremond et al., 2022; Wu et al., 2022a), the fine-grained multi-dimensional analysis with $d_{\mathsf{Eig}}$ could pave a promising way towards precise risk stratification by validating well-established and proposing novel features with prognostic importance in complex medical images. This can be concretely supported with biologically interpretable visualization examples generated by perturbing the largest eigenvalue(s)/eigenvector(s) in a given dataset of interest. This approach could thus be used to control for inherent variance in existing data repositories and generate prototypical examples of disease states such as highly aggressive tumors in radiological or pathological time series.

In combination of comprehensive quantification and biologically meaningful visualization, $d_{\mathsf{Eig}}$ thus adds a valuable tool for future work in the natural and medical image domains.

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
