# OpenReview forum: "Sorted eigenvalue comparison $d_{\mathsf{Eig}}$: A simple alternative to $d_{\mathsf{FID}}$"
_ICLR.cc/2023/Conference — Submitted to ICLR 2023_

### Official Review · Reviewer_6Rmw · 2022-10-21

**Confidence:** 4
**Correctness:** 2
**Technical Novelty And Significance:** 2
**Empirical Novelty And Significance:** 1
**Recommendation:** 3

**Clarity, Quality, Novelty And Reproducibility:**

The ideas in this work are presented clearly. The writing and overall quality
of the argument for (1) why the proposed method for computing $d_{FID}$ via
eigenvalues is universally beneficial and (2) the use of their new $d_{Eig}$
distance could be strengthened both with better experiments and more
theoretical justification. The experiments and results are reproducible.

**Strength And Weaknesses:**

**Strengths:**
- The Frechet inception distance (FID) is widely used.
  Heusel et al., (NeurIPS 2017) proposed using FID for GANS, which has kicked off a massive amount of subsequent research on distances between distributions that are useful in practice.
- The derivation of $d_{Eig}$ as an alternative to $d_{FID}$ is a nice contribution and could be studied more deeply.

**Weaknesses:**
- When introducing FID, it would also be valuable to draw connections to the
  Wasserstein distance-based definition (and equivalently couplings).
- The runtime reductions of 25% and 90% need more context (e.g., does this come
  from an improve constant factor in the running time, or is it an artifact of
  your data/implementation/machine?).
- A more detailed discussion about the differences between `scipy.linalg.sqrtm`
  and `np.linalg.eigvals` under the hood is needed? Don't both compute the Schur decomposition?
- Paper organization: Introduce all notation in a single section. Splitting
  between the beginning of Section 2 and Section 3 is non-standard. I also
  recommend putting all empirical results at the end of the paper.
- Figure 2a is a good experiment, but I think these plots would be easier to
  parse if the dimension increased from left to right, and if the number of
  samples on the x-axis increased from left to right (i.e., reverse everything).
  Moreover, I think language like "Approximation quality" or "Numerical stability"
  would be more helpful than "Oracle comparison".
- [page 6] The paper could benefit from a couple more sentences between Theorem
  10 about why the stability condition in Bao et al. (2015) is reasonable.
- The paper could benefit from fewer experiments, where each
  experiment has a little more substance and deeper theoretical backing (or
  connections to experiments in other related works).

**Suggestions:**
- [page 1] The notation $\lambda^{(1)}_{j}$ would better differentiate $S_1$ and $S_2$.
- [page 2] Suggestion: "the proposed $d_{FID}$ section" --> use \Cref{} to give actual section.
- [page 2] It would be useful to explicitly say $S_{i} \in \mathbb{R}^{p \times p}$ to
  make the orientation of your vectors clear.
- [page 3] suggestion: Add citation for the uniqueness property in Theorem 2.
- [page 3] suggestion: Add proof of Lemma 4 to the appendix for completeness.
- [page 4] The notation $\lesssim O(\epsilon)$ is redundant/imprecise.
- [page 5] suggestion: Format the parenthesis in Eq (5) so that they're at
  least as tall as the sqrt symbols.
- [page 8] All of the tables should use tabular environments instead of Numbers
  screenshots.
- [page 9] suggestion: Discuss related works as last section of the
  introduction.
- [page 9] typo: "as an example(Fremond et al.,...)"

**Summary Of The Paper:**

This work studies variations on the Frechet Inception Distance $d_{FID}$.
First, the authors show how we can implement between two datasets by using
more numerically stable eigenvalue subroutines instead of a full
matrix square root method. This leverages the fact that the covariance matrices
for the two sets of samples is PSD. This observation leads to practical running
time improvements. Second, the authors then propose the "Sorted eigenvalue
comparison" distance, which is motivated by FID when the two covariance matrices
are diagonalized by the same orthgonal matrix $Q$. The remainder of this work
explore the implications of using $d_{Eig}$ more broadly.

**Summary Of The Review:**

This work studies a clean mathematical distances and begins to explore (1) how
it can be computed faster in practice and (2) variants that also fast to
compute while being theoretically sound. I do not recommend that it be accepted
to ICLR 2023 given the overall quality of the paper, arguments, and experiments.
The work is interesting, but it needs to be developed quite a bit further
before being accepted to a conference of this caliber.

---

### Official Review · Reviewer_7tys · 2022-10-23

**Confidence:** 4
**Clarity, Quality, Novelty And Reproducibility:** Look good.
**Correctness:** 4
**Technical Novelty And Significance:** 2
**Empirical Novelty And Significance:** 3
**Recommendation:** 5

**Strength And Weaknesses:**

Before I post my summary, I want to suggest the authors to flip the orientation of the x-axis. It took me several minutes until I realize that the x-axis increases along the left direction. Normally, you would expect the number of samples to increase along the right direction, and the opposite is really confusing.

The main strength of the paper is the experimental evaluation where the authors demonstrate benefits of the eigenvalue-based formulation. In particular, d_{eig} can perform well with fewer data samples than d_{fid}. Nonetheless, my main concern is that a lot of the things discussed in the paper are straightforward. For example, there is no need to compute the eigenvalues of S_1S_2 using a non-symmetric eigenvalue solver (this is claimed as a drawback). This seems to be one of the main reasons why the proposed d_{eig} does better; but when S_1 is SPD, the pencil (S2,S_1^{-1}) is also SPD, and there is no need to use a non-symmetric eigenvalue solver.
Moreover, computing the square root of a matrix product is asymptotically equivalent to that of an eigenvalue decomposition, although I do agree that computing eigenvalues seems cheaper indeed; especially if someone uses the definition of d_{fid} in (1) which requires way too many floating-point operations. Section 2.2.2 holds when the two covariance matrices commute. Is this happening in practice? If not, the definitions in 2.2.2 become heuristics. Are these heuristics the ones reported in the experiments? I can tell that this seems to be the case in Section 3.2, in which case the authors do not compute the exact FID.

Overall, I think the paper is well-written and clear. While I support the idea, many of the modifications the authors present are relatively incremental (for example, (2) is well known in the literature).



**Summary Of The Paper:**

This paper presents two formulas to compute the Fréchet Inception Distance (FID) between real/generated datasets.
The authors provide some worst-case error analysis coupled with practical details and eigenvalue sorting. Numerical
experiments demonstrate the effectiveness of the proposed schemes.

**Summary Of The Review:**

See my answer to "Strength and weaknesses".

---

### Official Review · Reviewer_RbUs · 2022-10-24

**Confidence:** 4
**Correctness:** 3
**Technical Novelty And Significance:** 1
**Empirical Novelty And Significance:** 2
**Recommendation:** 3

**Clarity, Quality, Novelty And Reproducibility:**

The paper is clear and original. The paper doesn't seem novel. The experiments are sound.

**Strength And Weaknesses:**

Strength: $d_{Eig}$ is much cheaper to compute than $d_{FID}$.
Weakness:
1. The trick to accelerate $d_{FID}$ seems not hard.
2. Comparing to upper bounds. (See Summary of Review)
3. $d_{FIG}$ in experiment isn't more stable and it lacks of some desired properties.


**Summary Of The Paper:**

The authors study two similiarty distances for distribution shift, $d_{FID}$ (Frechet Inception Distance) and the prposed $d_{Eig}$ (Sorted Eigenvalue Comparison). The authors first showed by slightly modifying the $d_{FID}$ algorithm, the computational time for $d_{FID}$ can be improved. Then, the authors provide better upper bounds for this method. Finally, inspired by this algorithm, the authors propose $d_{Eig}$ and justify its stability by eigenvalue rigidity.

**Summary Of The Review:**

The paper is clear to follow, but however the main contribution of the paper is unclear. The first improvement trick seems to be not that hard, I found it hard to believe that no one has ever implemented in such way to save computational time--it would be more convincing that some well used library has been using $np.trace(sqrtm(S_1S_2))$ as an evidence. The second point: by comparing to upper bounds, it is not enough to say that one algorithm is better than the other; you need a lower bound or PAC-type of result to justify that. Finally, despite arguing $d_{Eig}$ is more stable alternative than $d_{FID}$, the empirical studies didn't show this. (Say in Fig 3, if you look at the relative error in the table, $d_{FID}$ has the smallest errors)

I agree with the authors that $d_{Eig}$ seems to be computationally cheaper, but however it seems to lack the desired mathematical properties in evaluation. As the contribution of the paper isn't clear for me, I tend to reject the paper for ICLR.

---

### Official Review · Reviewer_28R5 · 2022-10-27

**Confidence:** 3
**Correctness:** 3
**Technical Novelty And Significance:** 2
**Empirical Novelty And Significance:** 2
**Recommendation:** 5

**Clarity, Quality, Novelty And Reproducibility:**

**Clarity**: the paper is in general clearly written.

**Quality and Novelty**: the contribution seems to be of limited interest, some efforts are needed.

**Reproducibility**: no code for proof, but those may not be necessary? In any case, that depends on the precise contribution of this work.

**Strength And Weaknesses:**

**Strength**: the paper is in general clearly written. And the problem under study is of significance.

**Weaknesses**: some efforts are needed to better highlight the contribution of this paper. See my detailed comments below.

**Summary Of The Paper:**

In this paper, the authors proposed an improved Fréchet Inception Distance estimator (d_FID), as well as a sorted eigenvalue distance estimator (d_Eig), between two sample covariance matrices (SCMs).
Computational error bounds were established for these two methods, with a huge saving in the running time.
The authors then provided a few (statistical) arguments from random matrix theory to characterize the behavior of the largest eigenvalues of SCMs in Section 3.1.
The proposed d_Eig is then applied, as a simple alternative to d_FID, in Section 3.2, as the scores of GANs.


**Summary Of The Review:**

Detailed comments:

* some derivations in Section 2 are somewhat elementary. These are not enough to be the major contribution of the paper.
* Section 2.2.1: error bound of eigvals(): in the first sentence of this paragraph, s_j is not yet defined? or perhaps I missed something？
* Section 2.2.2 considers a very special case: I am wondering if there exists any practical situation where such a special case holds.
* Theorem 10 holds under the stability condition (Bao et al., 2015, Condition 1.1 (iii)) which should be stated explicitly. Also, it remains unclear whether Theorem 10 is a novel result.
* I do not understand why it is of interest to consider the "spike" behavior in Sec 3.1, and in which context.
* some numerical results, e.g., in Figure 2(c) and in Figure 3 are hardly visible: Not sure if they should be put in the appendix.
* in Section 3.2, the concrete application in GAN is considered. However, it remains unclear how the proposed improvement is important from a GAN perspective.
* No proof is given: this makes me feel that the results presented in the paper are all existing results. And no code is released for the numerical experiments.

---

### Decision · Program_Chairs · 2023-01-20

**Decision:**

Reject

**Justification For Why Not Higher Score:**

The reviewers thought the results were a bit incremental and the novelty may not be quite high enough for ICLR. Also, there was no author rebuttal to contest this claim.

**Justification For Why Not Lower Score:**

N/A

**Metareview: Summary, Strengths And Weaknesses:**

The reviewers thought the results were a bit incremental and the novelty may not be quite high enough for ICLR. Also, there was no author rebuttal to contest this claim.